# Characterization of the Hoof Bacterial Communities of Active Digital Dermatitis Lesions in Feedlot Cattle

**DOI:** 10.3390/microorganisms12071470

**Published:** 2024-07-19

**Authors:** Nicholas S. T. Wong, Nilusha Malmuthuge, Désirée Gellatly, Wiolene M. Nordi, Trevor W. Alexander, Rodrigo Ortega-Polo, Eugene Janzen, Murray Jelinski, Karen Schwartzkopf-Genswein

**Affiliations:** 1Large Animal Clinical Sciences, Western College of Veterinary Medicine, University of Saskatchewan, Saskatoon, SK S7N 5C9, Canada; nicholas.wong@telusagcg.com (N.S.T.W.); karen.genswein@agr.gc.ca (K.S.-G.); 2Lethbridge Research and Development Centre, Agriculture and Agri-Food Canada, Lethbridge, AB T1J 4B1, Canada; nilusha.malmuthuge@agr.gc.ca (N.M.); trevor.alexander@agr.gc.ca (T.W.A.); rodrigo.ortegapolo@agr.gc.ca (R.O.-P.); 3Technology Access Centre for Livestock, Olds College of Agriculture & Technology, Olds, AB T4H 1R6, Canada; 4Faculty of Veterinary Medicine, University of Calgary, Calgary, AB T2N 4N1, Canada; janzene@ucalgary.ca

**Keywords:** digital dermatitis, active lesions, 16S amplicon sequencing

## Abstract

Digital dermatitis (DD) is a costly hoof infection, causing lameness and pain in feedlot cattle. DD lesions can develop nonlinearly through a series of clinical stages, which can be classified by Dopfer’s M-stage scoring system. This widely adopted lesion scoring system recognizes five DD stages, where M1 (early lesion), M2 (acute ulcerative lesion), and M4.1 (chronic proliferative lesion with new developing lesion) are considered active but separate stages of the disease. This study assessed the skin surface microbiota of the active DD lesions of feedlot cattle. The DD lesions from three commercial feedlots were swabbed and then scored according to Dopfer’s M-stage scoring system. Swab samples were collected from 12 M2- and 15 M4.1-stage lesions. A total of 21 control swab samples from healthy contralateral feet (DD control) were classified as stage M0. An additional six skin swabs (M0) were collected from completely healthy (CH control) cattle with no lesions. The bacterial communities of active DD lesions (M2 and M4.1) and healthy skin (M0) were profiled using 16S amplicon sequencing. Diversity analyses showed that the hoof bacterial communities of M2 and M4.1 lesions were each distinct from those of M0 skin. However, the bacterial communities between the two active lesion stages were not different from each other. A significant increase in the relative abundance of *Spirochaetota* and *Fusobacteriota* and an overall decrease in bacterial diversity contributed to the altered bacterial communities in M2 and M4.1 lesions compared to those of healthy skin (M0). Although stages M2 and M4.1 are considered clinically different stages, the lesion-associated bacterial community is similar between the two active stages.

## 1. Introduction

Digital dermatitis (DD) has become a major cause of lameness in feedlot cattle, incurring significant economic costs associated with treatment and reduced production performance [1]. DD is also a welfare concern due to the pain associated with the ulcerative stages of the lesion [2]. The manifestation and microbial composition of DD lesions are similar in dairy and feedlot cattle, with lesions typically located on the skin of the plantar surface between the heel bulbs [3,4]. While results vary between studies, the consensus is that DD has a polymicrobial etiology [5]. Furthermore, *Treponema* species are considered the main taxonomic group associated with DD in both dairy and beef lesions [5,6]. Beninger et al. [7] found that the absolute abundance and the number of unique *Treponema* species were higher in the active and ulcerative lesion stages than in the chronic stages, healing, or healthy skin of dairy cattle [7]. As a possible biomarker of chronic stages, *Mycoplasma* was suggested as an important bacterial genus during the transition between lesion stages [8]. In addition, other genera such as *Fusobacterium*, *Porphyromonas*, *Bacteroides*, *Mycoplasma*, *Campylobacter*, and *Dichelobacter* have been associated with DD [8,9,10,11,12].

First described by Dopfer et al. (1997), the M-stage scoring system is a widely adopted classification scheme for describing the stages of DD [13]. The system was modified and adapted by Berry et al. (2012) [14] by classifying DD lesions into five morphological or clinical stages: (1) M0, normal skin without any lesions; (2) M1, an early active lesion with a diameter of <2 cm; (3) M2, a highly ulcerative lesion with a diameter of >2 cm characterized by a bright red, strawberry-like appearance and is painful; (4) M3, a healed stage of a nonpainful lesion characterized by a black scab after treatment; (5) M4, the chronic, inactive, and nonpainful stage of a lesion usually characterized by hyperkeratosis and proliferation, which has a gray, wart-like appearance and/or hair-like projections; and (6) M4.1, a chronic and active stage where an active M1 or M2 lesion develops within the perimeter of an existing chronic M4 lesion [13,14]. 

Clinical staging plays an important role in devising effective treatment strategies. It is assumed that chronic lesions act as a disease reservoir, which complicates treatment and control measures [15]. As a primary treatment option, topical antimicrobials such as oxytetracycline can be effective in reducing the bacterial burden of *T. phagedenis* and *T. pedis* in active DD lesions [16]. Nonetheless, existing treatment options including topical products have mixed results in achieving successful outcomes [10,17]. DD relapses are common, with one study reporting a relapse rate of 32% in dairy cattle [10,18]. When DD lesions do not completely heal after treatment, more aggressive and prolonged treatments may be required [18]. Copper or zinc sulfate footbaths are commonly used to control DD in dairy herds [2]. However, the existing control and treatment protocols that include footbaths have been ineffective in controlling DD [6,18]. Logistically, footbaths are also more challenging in feedlots because the cattle are handled infrequently, and the footbaths can freeze during the winter months. Added to these challenges are environmental concerns regarding the disposal of large quantities of footbath solution. 

Multiple studies involving the biopsies of DD lesions have shown that the microbiota changes with disease progression [7,8,11]. However, invasive sampling procedures such as punch biopsies are not a practical option in commercial feedlots. Surface swabs, however, are minimally invasive and have been used previously to describe the microbiota of DD lesions [19]. Therefore, understanding the transition of the surface microbiota from an active, ulcerative lesion to a chronic lesion may provide insight into DD lesion dynamics. We previously reported that DD lesions have significantly different bacterial compositions than healthy skin [19], but whether bacterial communities differ between active M stages has not been investigated, particularly at the surface community level. Therefore, there may be further diagnostic potential for swabs to not only detect presence of DD but also to determine disease stages based on surface communities, which may in turn provide useful insight into effective topical treatments for the specific active stages. The objective of this study was to characterize the skin surface microbiota of active DD lesions (M2 and M4.1).

## 2. Materials and Methods

Sample Collection: From January 2019 to March 2020, weekly visits were made to three commercial feedlots (>20,000 head capacity) in southern Alberta, Canada, by researchers experienced in identifying DD and other hoof lesions. On the morning of each visit, feedlot staff placed lame cattle in a holding pen. The degree of lameness was scored using the Step Up^®^ beef locomotion scoring system (Zinpro, Eden Prairie, MN, USA): 0—normal gait; 1—initial signs of lameness, no obvious sign of limp; 2—obvious limp, slight head bob, and arched back; 3—animal applies little to no weight on the affected limb and may be reluctant and unable to move. Each lame animal was restrained in a squeeze chute, and the affected hooves were examined for DD lesions. Lesion stage was classified according to the modified Dopfer M-stage scoring system [13,14]. 

DD lesions were sampled by rotating a sterile cotton-tipped applicator swab (Puritan Medical Products Company, Guilford, ME, USA) across the active and chronic parts of the lesion. Each sample was paired to a corresponding control swab from the healthy skin of the contralateral hoof (DD control). In addition, completely health control samples (CH control) were obtained from the interdigital and plantar skin regions of healthy cattle that were free of hoof lesions. Swabs were placed into sterile tubes and transported on ice until storage at −80 °C. In total, 54 hoof samples were obtained: 27 M0 (21 DD control, 6 CH control), 15 M2, and 12 M4.1 (Figure 1; Table 1).

DNA Extraction: Genomic DNA was extracted from the cotton swabs using a Qiagen DNeasy Powerlyzer/Powersoil kit (Qiagen, Germantown, Maryland, USA), according to the manufacturer’s instructions. DNA concentration was measured using a Qubit dsDNA Quantification HS (High Sensitivity) Assay kit (ThermoFisher Scientific, Waltham, MA, USA), and quality was checked using 1% agarose gel electrophoresis. Extracted samples were stored at −20 °C. Samples were diluted to 10 ng/µL and transferred onto 96-well plates to prepare amplicons for sequencing. For samples below the 10 ng/µL concentration threshold, a maximum 10 µL volume was used to prepare amplicons. 

The 16S rRNA Amplicon Sequencing: Library preparation and sequencing were performed by Genome Quebec (Montreal, QC, Canada). The V4 hypervariable region of the 16S rRNA gene was amplified with the primer set 515F (5′–ACACTGACGACATGGTTCTACAGTGCCAGCMGCCGCGGTAA-3′)/806R (3′–TACGGTAGCAGAGACTTGGTCTGGACTACHVGGGTWTCTAAT-5′) [20]. Briefly, PCR amplification was performed using Roche FastStart master mix under the following conditions: 94 °C for 2 min, followed by 26 cycles of 94 °C for 30 s, 58 °C for 30 s, 72 °C for 7 min, and a final elongation step at 72 °C for 7 min. Barcoding and indexing were conducted by using in-house PCR conditions at Genome Quebec. Verification of barcode incorporation for each sample was performed on 2% agarose gel. Quantification of each amplicon was conducted with a Quant-iT™ PicoGreen^®^ dsDNA Assay Kit (Life Technologies, Carlsbad, CA, USA). A sequencing library was generated by pooling the same quantity (ng) of each amplicon. Following clean up, the sequencing library was quantified using Kapa Illumina GA with a Revised Primers-SYBR Fast Universal kit (Kapa Biosystems Inc., Wilmington, MA, USA), and average fragment size was determined using a LabChip GX (PerkinElmer, Waltham, Massachusetts, USA) instrument. Sequencing was performed with a MiSeq Reagent Kit v2 500 cycles from Illumina (2 × 250 bp) with LNATM-modified custom primers.

Quality control was performed and sequence data analysis of Raw FASTQ files were downloaded and imported to QIIME2 version 2021.4 [21] in paired-end FASTQ Phred 33 format. Denoising was performed using DADA2 in QIIME2 to merge paired reads, filter and denoise sequences, and remove chimeric sequences. Based on the Phred score of the sequences, the truncation length was set at 240 base pairs for both the forward and reverse reads [22]. Representative sequences and an amplicon sequence variant (ASV) table were generated after the original data were filtered through the DADA2 pipeline [22]. The align-to-tree-MAFFT-FastTtree plug-in was used to align representative sequences and construct a phylogenetic tree [23,24]. A feature classifier, “classify-sklearn”, was used to assign taxonomic ranks using the Silva database version 138 classifier trained for the V4 hypervariable region of the 16S rRNA gene [25,26]. Extraction blanks (negative control) went through the same preliminary quality control (QC) as the samples. Following denoising, the negative control only had an average of 267 sequences, being significantly fewer than the hoof sample with the lowest number of sequences (4371). Thus, potential contaminations due to reagents, PCR, and sequencing library preparation accounted for <2% of all sequences. Filter featuring was performed to remove mitochondria, chloroplasts, and archaea from the ASV table. In addition, unassigned ASVs (e.g., taxonomic assignment only at Kingdom: Bacteria, Kingdom: Unassigned) were removed using a feature filtering option. Only genera that appeared in more than 10% of all samples and appeared in more than 10% of the total frequency were kept for differential abundance analysis, phylogenetic analysis, and taxonomic bar plots. Therefore, downstream analyses were performed using only the sequences generated from the hoof swabs. See Appendix A for the quality control test results.

Diversity Analyses: The alpha-rarefaction plug-in was used to generate rarefaction curves for each individual sample, using a minimum sampling depth of 4366 [21]. The core-metric-phylogenetic plugin generated alpha and beta diversity metrics [21]. The Shannon and Chao1 indices were included as alpha diversity measures to assess the richness and evenness within bacterial communities generated from the active DD lesions and control skin samples [27,28]. Additionally, alpha-diversity analyses were performed for each stage of lesion for each phylum that was found in more than 50% of the samples and more than 10% of the total frequency using separate minimum sampling depth. The alpha-group-significance plugin was used to perform a Kruskal–Wallis, nonparametric test with a false-discovery rate (FDR) correction or a Benjamini–Hochberg adjusted *p*-value [29,30].

Beta diversity analysis was performed using the weighted Unifrac distance metric, and principal coordinates analysis (PCoA) plots were generated using the EMPeror tool to visualize the dissimilarity between bacterial communities [31,32]. The beta-group-significance and adonis plugin were used to assess the significant differences between bacterial communities using a permutational multivariate analysis of variance (PERMANOVA) non parametric test [33]. Post hoc pairwise comparisons were performed between a lesion type and its corresponding control skin. Adjusted *p*-values were generated using the Benjamini–Hochberg procedure (1995) [30]. 

Phylogenetic Analysis: The align-to-tree-MAFFT-FastTree pipeline, using representative sequences as input, was used to generate a phylogenetic tree [23,24]. The phylogenetic tree was visualized via the plug-in empress community and edited using the EMPress tool [34]. The inner colored shades highlight the bacterial phyla (inner ring) representative of the phylogenetic tree. Relative abundance bar plots (on the outer ring) for each M stage were further annotated for the strains contained within each phylum.

Taxonomic Summary: The taxa barplot plug-in generated taxonomic bar plots on a filtered table collapsed at the phylum level (level 2) [21]. These bar plots show the relative abundance (%) of the 10 most prevalent phyla across all samples. Each bar represents an individual hoof sample separated by M stage. In addition, relative abundance box plots were created for the top 10 most prevalent bacterial genera across all samples, where the relative abundance for each genus was grouped by M stage. A Kruskal–Wallis test was used to test the significant differences in a phylum’s relative abundance between M stages [29]. Post hoc pairwise comparisons between M stages were made with a Benjamini–Hochberg adjusted *p*-value [30].

Differential Abundance Analysis: The ALDEx2 q2-plugin was used to assess differentially abundant (DA) bacterial genera between the lesion and control skin [35]. Two pairwise analyses were performed separately for each active lesion stage (M2 and M4.1) and compared to the normal (control) skin (M0) at the phylum and genus levels. Within the ALDEx2 analysis, centered log ratios (CLRs) of bacterial genera and Welch’s *t*-test were used to identify DA bacterial genera. Adjusted *p*-values were calculated using the Benjamini–Hochberg procedure, and DA genera were declared at an adjusted *p*-value ≤ 0.05 and an effect size ≥0.8 or ≤−0.8. A positive effect size signified that a genus was more abundant in the lesion than in the control skin, whereas a negative effect size signified a genus was more abundant in control than lesion skin. 

The Songbird q2-plug-in was used to explore the links between bacterial taxa and lesion types [36]. It utilizes a multinomial regression approach to provide information on the relative association of bacteria with a given covariate. The parameters used in the Songbird analysis included the metadata columns DD M stage: --p-formula “C (DDMStage, Treatment (‘M0))” --p-epochs 10,000 --p-differential-prior 0.5 --p-summary-interval 1. When the treatment model was compared to a “null” model with a --p-formula of 1, a positive pseudo-Q2 value indicated that the model was not overfitted. M-stage-specific genera were identified by sorting the genera associated with each stage in the Songbird output from the highest to lowest.

## 3. Results

### 3.1. Diversity Analyses

The bacterial communities of M2 and M4.1 lesions were less diverse than those of the control (M0) skin (Chao1 and Shannon metrics, adj-*p* < 0.05; Figure 2). The least diverse bacterial community (adj-*p* < 0.01) was observed in M2 lesions. The Shannon index (richness and evenness) was lower in the bacterial communities generated from stage M2 than those of stage M4.1 (Figure 2B). When the alpha diversity within phyla was compared, the diversities of *Bacillota*, *Bacteriodota*, and *Actinomycetota* were lower (adj-*p* < 0.05) in the bacterial profiles generated from stage M2 than those from stage M0 (Figure 3A–F). Compared to stage M0, the diversity of *Proteobacteria* was lower (adj-*p* < 0.01) in M2 and M4.1 lesions (Figure 3G,H), while that of *Spirochaetota* was higher (adj-*p* < 0.01) in active lesions (Figure 3I,J). The Shannon index of *Bacillota* and Chao1 index of Proteobacteria differed (adj-*p* < 0.05) (Figure 3B–G) between active lesions.

Feedlot accounted for 13.6% of the observed variation in the hoof bacterial communities (PERMANOVA-R2 = 0.136, F = 4.00, *p* < 0.01; Figure 4A). The presence or absence of a lesion (regardless of the staging) contributed to 30.1% of the variations observed between bacterial communities (Figure 4B). Bacterial community structure was also affected by disease state (M0, M2, M4.1), contributing to 32.2% of the variation among the bacterial communities (PERMANOVA-R2 = 0.322, F = 12.1, *p* < 0.01; Figure 4C). The bacterial profiles generated from M0 samples clustered apart from M2 and M4.1 (Figure 4C; Table 2). However, there was no separation between the M4.1 and M2 bacterial communities (Appendix A; Table 2).

### 3.2. Phylogenetic Analysis

The phylogenetic tree showed that most of the bacterial taxa from phyla *Bacillota*, *Bacteriodota*, *Actinomycetota*, and *Proteobacteria* were shared among M stages (Figure 5). However, stages M2 and M4.1 showed two specific clusters that contained taxa from *Spirochaetota* and *Fusobacteriota* (as indicated by the black arrows) (Figure 5).

### 3.3. Taxonomic Summary

The predominant bacteria identified from the hoof swabs were *Bacillota, Bacteroidota*, and *Proteobacteria*, regardless of the stage, with a higher individual variation (Figure 6). The relative abundances of *Proteobacteria* and *Actinomycetota* were significantly higher in M0 samples than in M2 and M4.1 lesions (adj-*p* < 0.01), while those of *Spirochaetota*, *Fusobacteriota*, and *Campilobacterota* were significantly higher in both active lesions than in M0 (adj-*p* < 0.01) (Figure 6). At the genus level, *Treponema* and *Prophyromonas* were numerically higher across active lesions than in M0 skin (Figure 7).

### 3.4. Differentially Abundant (DA) Analysis

In the M0- and M2-stage pairwise comparisons, 4 phyla and 13 genera were differentially abundant, while 6 phyla and 17 genera were differentially abundant between M0 and stage M4.1 (Table 3). There were no DA taxa between M2 and M4.1 stages. Songbird analysis revealed that 11 out of top 15 taxa (*Acholeplasma*, *Amnipila*, *Catonella*, *Falsiporphyromonas*, *Filifactor*, *Fretibacterium*, *Lachnospiraceae_AC2044_group*, *Parvimonas*, *Peptostreptococcaceae*, *Roseburia*, and *Treponema*) were associated with M2 and M4.1 (Table 4).

## 4. Discussion

The bacterial profiles generated using skin swabs revealed that the presence of a DD lesion accounted for 30.1% of the variation in the hoof bacterial community, while the feedlot from which the DD animal was reared only accounted for 13.6% of the variation. Although the skin surface microbial communities of active DD lesions (M2 and M4.1) were significantly different from those of healthy skin (M0), there were no differences between those of M2 and M4.1 lesions. Krull et al. (2014) also reported that DD stage (based on Iowa DD scoring) contributed to 54% of the variation observed among the bacterial profiles generated from biopsy samples [8]. Compared to the M-stage scoring system, the Iowa DD scoring system includes more early-stage lesions and does not include a stage with active and chronic components. Furthermore, their PCoA based on 16S amplicon sequencing showed a separation of bacterial profiles generated using biopsies collected at stages 3 and 4 apart from stage 0, 1, and 5 [8]. Additionally, the authors reported that the metagenome-based microbial profiles were different among stages, except between stage 0 (control biopsy) and stage 5 (post-treatment biopsy). However, the skin swabs based 16S profiles did not differ between the active stages. These differences could be due to the variations in sequencing techniques, number of samples, and the DD stages selected for the analysis.

Although the DD lesions in our study were classified as distinct M stages, in both stages, *Spirochaetota* and *Fusobacteriota* were the primary contributors to the microbial dysbiosis, whereas these phyla were largely absent in M0 skin. There was notable increases in both the richness as well as in the relative abundance of *Spirochaetota* in the DD lesion versus healthy skin bacterial communities. Multiple studies have reported an increase in the abundance of *Treponema* species with DD lesions [3,7,8,11]. Likewise, the strong association of *Fusobacterium* and *Treponema* with M2- and M4.1-stage lesions in the present study is similar to the differential abundance findings in the Caddey et al. (2021s DD biopsy-based beef study, which also used a modified M-stage scoring system [4]. Krull et al. (2014) [8] also reported that an increased abundance of *Treponema* is a biomarker for type 3 (acute) and type 4 (chronic) lesions according to the Iowa scoring system, indicating that *Treponema* (or phylum *Spirochaetota*) is a major driver of DD lesions, and skin swabs can capture the dynamics of hoof bacterial communities during lameness lesion development. The consistency in findings is significant because we swabbed the surface of the skin, whereas others used the more invasive and time-consuming procedure of taking skin biopsies. It has been suggested that *Mycoplasma* may play a role in transitioning from chronic to active stages [4,8]. However, this potential role that *Mycoplasma* plays in chronic lesions is not clear based on our differential abundance analysis, where its abundance was not significantly different between M2- and M4.1-stage lesions. 

Our findings on differentially abundant hoof bacterial taxa are consistent with those of previous studies, wherein *Treponema* [3,4,8,9,11,37,38,39], *Mycoplasma* [4,8,9,11,37,38,39] *Prophyromonas* [4,8,9,37,38], *Fusobacterium* [4,8,11,27], *Peptostreptococcacae* [4,27] *Campylobacter* [4,8,9,11,27], *Murdochiella* [4,39], *Acholeplasma* [9,38,39], *Catonella* [4,9,37], *Fretibacterium* [4], and *Parvimonas* [39] have been associated with DD lesions. Of interest is that a number of M2- and M4.- associated taxa are known to persist in the oral cavities of animals and humans. These commonly shared taxa include *Fusobacterium* [40,41,42], *Catonella* [43], *Filifactor* [44], *Fretibacterium* [41,45], *Peptostreptococcus* [42,46], *Mycoplasma* [47], and *Absconditabacteriales*_(SR1) [41]. It can be speculated that these commensal genera become opportunistic pathogens that contribute to the inflammatory process [6,38]. 

The lack of M1, M3, and M4 samples is a study limitation, but it reflects the reality that early (M1) and inactive (M3, M4) DD lesions can be difficult to detect and diagnose in feedlot operations [13]. Since the main criterion for enrolled animals was lameness, animals that were affected with DD tended to have painful ulcerative lesions (M2) or advanced chronic lesions (M4.1). This bias in lesions was compounded by our choice of using a cross-sectional study design, which provided a snapshot of the lesion community at a single point in time. Ideally, the lesions would have been swabbed multiple times over the progression of DD; however, this was not logistically possible in the commercial feedlots that participated in this study. Future research should explore the physical environment, such as the soil, in which DD-infected cattle live to further investigate the association with the DD microbiota. In addition to including more M stages, future swab-based studies should adopt a longitudinal design that follows DD lesions in the same animals in order to capture surface DD lesion dynamics more fully in feedlot cattle.

## 5. Conclusions

Underlying chronic lesions (M4.1) did not seem to influence the surface bacterial taxa composition of active lesions, which suggests that similar taxa may be driving the active disease process, in at least active M stages. The findings suggest *Spirochaetota* and *Fusobacteriota* were the leading taxa driving dysbiosis. The similar bacterial taxa found in DD lesions among the present study and the literature support the use of surface swabbing as a sampling method, which can facilitate less invasive, faster, and more efficient sample collection. As lesion staging can be important in determining treatment plans, future swab-based studies can further investigate the virulence factors and antimicrobial resistance of the surface bacterial communities of different stages of DD lesions. 

## Figures and Tables

**Figure 1 microorganisms-12-01470-f001:**
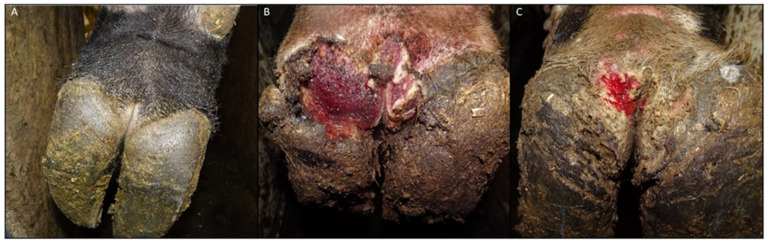
DD M stages included in this study. (**A**) A healthy foot (stage M0). (**B**) An M2-stage lesion, characterized by >2 cm ulcerative, bright red, and raised appearance. (**C**) An M4.1-stage lesion, characterized by a chronic lesion that may have a thickened, proliferative appearance along with a smaller active M1 or M2 lesion.

**Figure 2 microorganisms-12-01470-f002:**
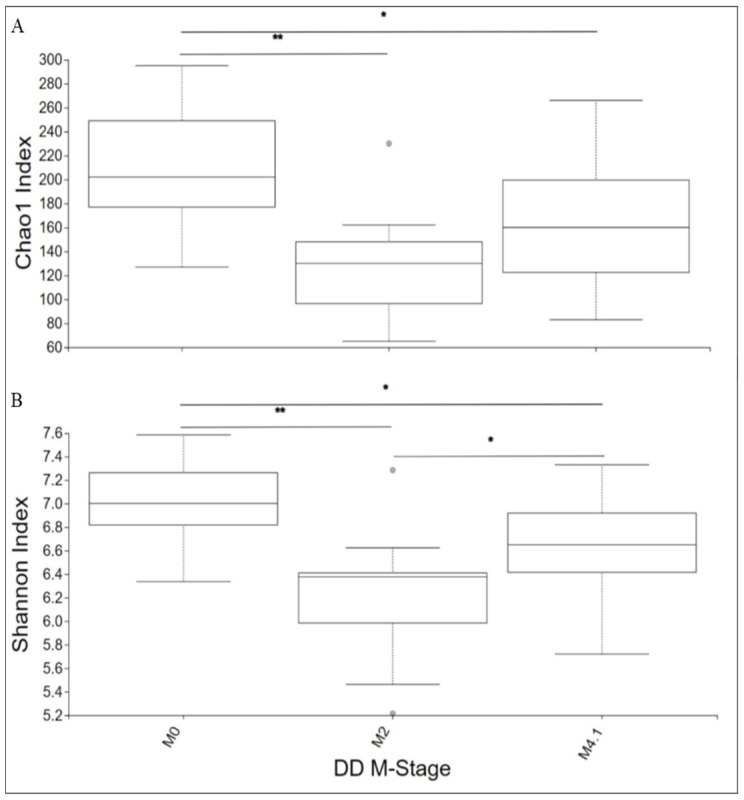
Boxplots of Chao1 diversity index (**A**) and Shannon diversity index (**B**) as a measure of overall alpha diversity for each M-stage group: * indicates a corrected *p*-value between 0.01 and 0.05 when a significant difference was detected; ** indicates a corrected *p*-value < 0.01 when a significant difference was detected. Pairwise analysis compared the alpha diversity indices of M2/M4.1 lesions and M0 control skin.

**Figure 3 microorganisms-12-01470-f003:**
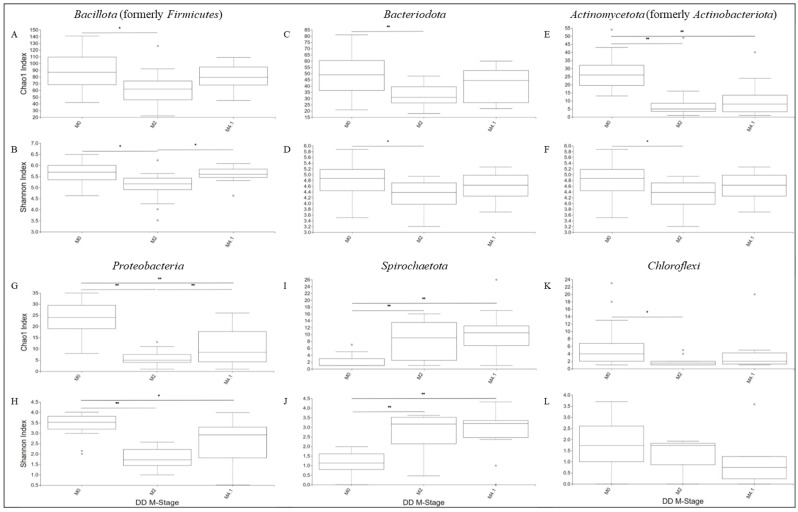
Boxplots of Chao1 diversity index and Shannon diversity index as a measure of Bacillota (**A**,**B**), Bacteriodota (**C**,**D**), Actinomycetota (**E**,**F**), Proteobacteria (**G**,**H**), Spirochaetota (**I**,**J**), and Chloroflexi alpha diversity (**K**,**L**) for each M-stage group: * indicates a corrected *p*-value between 0.01 and 0.05 when a significant difference was detected; ** indicates a corrected *p*-value < 0.01 when a significant difference was detected. Pairwise analysis compared the alpha diversity indices of M2/M4.1-lesion and M0-control skin.

**Figure 4 microorganisms-12-01470-f004:**
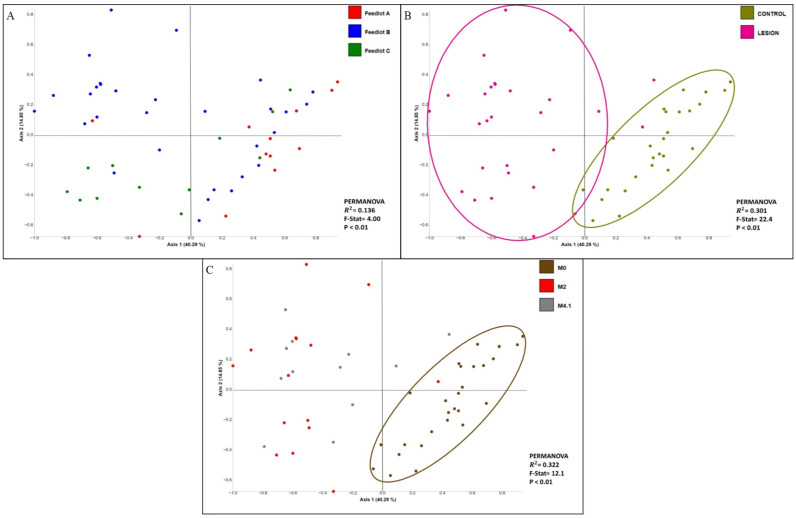
Principle coordinates analysis plot of weighted Unifrac distance metric generated from hoof bacterial communities (**A**) comparison of hoof bacterial profiles generated from different feedlots, (**B**) comparison of hoof bacterial profiles generated from different sample types (lesion vs. control), (**C**) comparison of hoof bacterial profiles generated from different M-stages.

**Figure 5 microorganisms-12-01470-f005:**
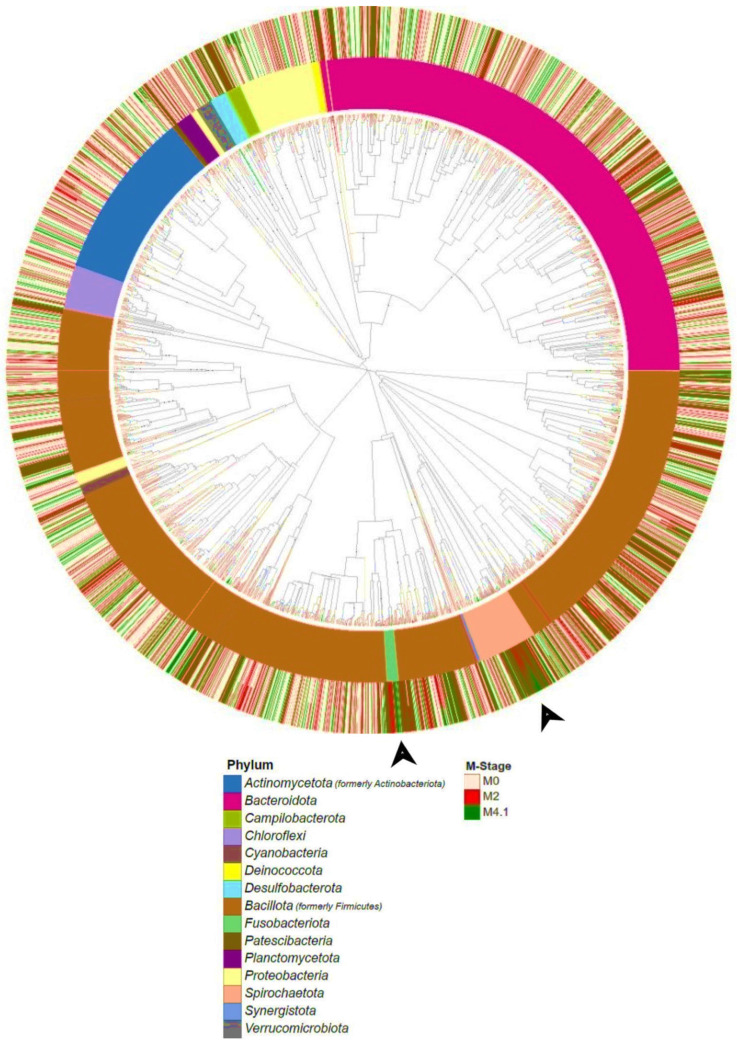
Phylogenetic tree based on 16S rRNA gene sequences (V4 hypervariable region) comparison from all DD control and DD lesion samples showing the relationship between M0 (control), M2 lesion, and M4.1 lesion samples at the phylum level. The inner ring represents the phyla to which the branch tips belong, while the outer ring represents the relative abundance boxplot.

**Figure 6 microorganisms-12-01470-f006:**
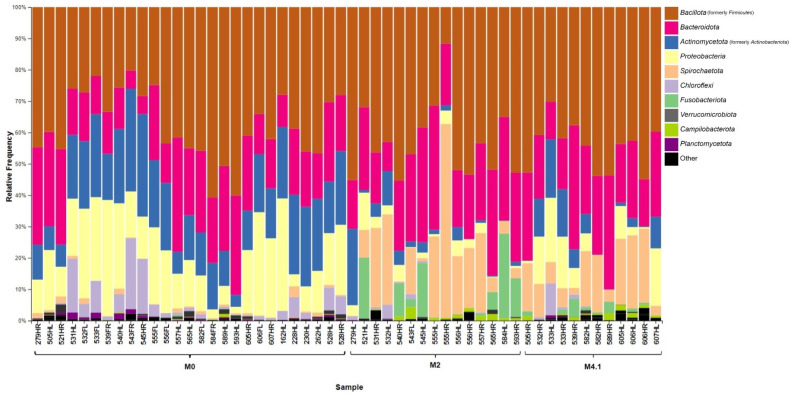
Taxonomic summaries of individual hoof samples showing the percent relative abundance of the top 10 bacterial phyla. Samples were grouped together by swab type and M stage.

**Figure 7 microorganisms-12-01470-f007:**
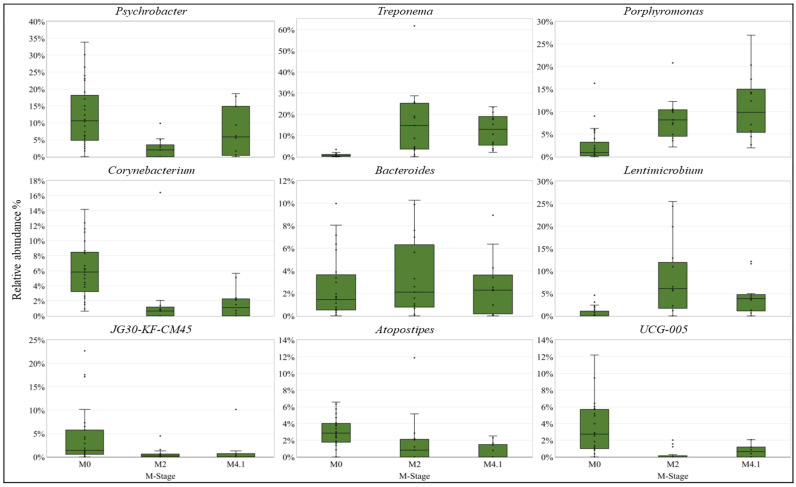
Box plots showing the percent relative abundance of nine most prevalent bacterial genera in each M stage.

**Table 1 microorganisms-12-01470-t001:** Distribution of samples across three feedlots by swab type and M stage.

	CH Controls	DD Controls	DD Lesions	
M Stage	M0	M0	M2	M4.1
Feedlot A	6	3	3	0
Feedlot B	0	12 *	8	10
Feedlot C	0	6	4	2
Total (*n* = 54)	6	21	15	12

* Three animals in feedlot B had DD lesions in two hoofs, which resulted in two DD lesion hooves and one control hoof samples for those animals.

**Table 2 microorganisms-12-01470-t002:** Beta diversity analysis of PERMANOVA pairwise comparison between M-stage sample groups based on weighted Unifrac distance metric.

Feedlot	Pairwise Comparison		PERMANOVA			
	Group 1	Group 2	Sample Size	F-Stat	R2	Adj *p*-Value *
All	M0	M2	42	18.2	0.313	<0.01
		M4.1	39	14.4	0.280	<0.01
	M2	M4.1	27	1.32	0.050	0.228
A	M0	M2	12	3.80	0.275	<0.01
B	M0	M2	20	15.0	0.455	<0.01
		M4.1	22	11.1	0.358	<0.01
	M2	M4.1	18	1.62	0.092	0.108
C	M0	M2	10	6.59	0.451	<0.01
		M4.1	8	3.74	0.383	0.037
	M2	M4.1	6	2.15	0.349	0.133

* Adj *p*-value: Benjamini–Hochberg adjusted *p*-value controlled for false-discovery rate. Significance at 0.05.

**Table 3 microorganisms-12-01470-t003:** ALDEx2 output of differentially abundant bacterial taxa associated with each M stage in pairwise comparisons between each M-stage lesion and M0 skin.

Taxon	M-Stage	Control CLR *	Lesion CLR *	Effect Size	Adj *p*-Value	Pairwise Comparison	Phylogenetic Level
*Spirochaetota*	M2	1.76	6.25	0.96	<0.01	M0 vs. M2	Phylum
*Fusobacteriota*	M2	−5.74	2.84	0.86	0.01	M0 vs. M2	Phylum
*Proteobacteria*	M0	6.90	4.19	−1.05	<0.01	M0 vs. M2	Phylum
*Actinomycetota*	M0	7.09	3.71	−1.18	<0.01	M0 vs. M2	Phylum
*Spirochaetota*	M4.1	1.76	5.77	1.39	<0.01	M0 vs. M4.1	Phylum
*Fusobacteriota*	M4.1	−5.76	1.24	1.12	<0.01	M0 vs. M4.1	Phylum
*Synergistota*	M4.1	−6.44	−0.63	0.86	0.03	M0 vs. M4.1	Phylum
*Planctomycetota*	M0	0.18	−5.04	−0.81	0.02	M0 vs. M4.1	Phylum
*Chloroflexi*	M0	3.55	−2.46	−0.93	<0.01	M0 vs. M4.1	Phylum
*Actinomycetota*	M0	7.08	2.81	−1.18	<0.01	M0 vs. M4.1	Phylum
*Porphyromonas*	M2	5.18	9.14	1.38	<0.01	M0 vs. M2	Genus
*Treponema*	M2	3.94	9.46	1.30	<0.01	M0 vs. M2	Genus
*Amnipila*	M2	−4.01	6.76	1.14	<0.01	M0 vs. M2	Genus
*Mycoplasma*	M2	−2.43	7.43	1.12	<0.01	M0 vs. M2	Genus
*Catonella*	M2	−4.01	5.86	1.11	<0.01	M0 vs. M2	Genus
*Peptostreptococcaceae*	M2	−0.55	6.28	1.10	<0.01	M0 vs. M2	Genus
*Fusobacterium*	M2	−3.57	5.91	0.95	0.02	M0 vs. M2	Genus
*Parvimonas*	M2	−4.23	0.95	0.89	0.02	M0 vs. M2	Genus
*Lentimicrobium*	M2	1.92	8.52	0.84	0.01	M0 vs. M2	Genus
*Falsiporphyromonas*	M2	−4.22	−0.04	0.83	0.03	M0 vs. M2	Genus
*Anaerovibrio*	M2	1.97	7.09	0.81	0.02	M0 vs. M2	Genus
*Jeotgalibaca*	M0	5.02	−2.13	−0.91	0.02	M0 vs. M2	Genus
*Olsenella*	M0	4.97	−0.81	−1.27	<0.01	M0 vs. M2	Genus
*Amnipila*	M4.1	−4.03	7.04	2.50	<0.01	M0 vs. M4.1	Genus
*Peptostreptococcaceae*	M4.1	−0.44	6.57	1.77	<0.01	M0 vs. M4.1	Genus
*Treponema*	M4.1	3.94	8.67	1.69	<0.01	M0 vs. M4.1	Genus
*Mycoplasma*	M4.1	−2.57	7.23	1.61	<0.01	M0 vs. M4.1	Genus
*Clostridia_vadinBB60_group*	M4.1	2.12	6.66	1.38	<0.01	M0 vs. M4.1	Genus
*Fusobacterium*	M4.1	−3.41	4.62	1.28	<0.01	M0 vs. M4.1	Genus
*Lachnospiraceae_AC2044_group*	M4.1	−4.04	5.12	1.24	<0.01	M0 vs. M4.1	Genus
*Roseburia*	M4.1	−3.00	5.86	1.23	<0.01	M0 vs. M4.1	Genus
*Murdochiella*	M4.1	−3.52	3.92	1.16	<0.01	M0 vs. M4.1	Genus
*Acholeplasma*	M4.1	−0.72	6.21	1.07	<0.01	M0 vs. M4.1	Genus
*Fretibacterium*	M4.1	−4.21	3.21	1.02	0.03	M0 vs. M4.1	Genus
*Porphyromonas*	M4.1	5.22	8.81	1.00	<0.01	M0 vs. M4.1	Genus
*Campylobacter*	M4.1	−0.87	5.36	0.91	0.01	M0 vs. M4.1	Genus
*Anaerovibrio*	M4.1	1.92	6.41	0.89	0.02	M0 vs. M4.1	Genus
*Filifactor*	M4.1	−3.97	2.14	0.88	0.04	M0 vs. M4.1	Genus
*Olsenella*	M0	4.98	1.97	−0.80	0.03	M0 vs. M4.1	Genus
*Clostridium_sensu_stricto_1*	M0	5.51	−0.18	−0.99	0.02	M0 vs. M4.1	Genus

* CLR—median value of a genus’s transformed abundance data in the lesion or control skin group of samples; a positive CLR indicates higher abundance, and negative CLR indicates lower abundance. ALDEx2 transforms the compositional abundance data into centered log ratios (CLRs) for genera in the lesion and control skin groups. It then compared them using a two-sample Welch’s *t*-test to determine significance. Only significant genera with adj *p*-value < 0.05 and effect size > 0.8 are shown.

**Table 4 microorganisms-12-01470-t004:** Songbird output of the top 15 taxa associated with each M stage of DD lesions. Single-underlined taxa were associated with both M-stage lesions.

Rank	M0 Associated Taxon	M2 Associated Taxon	M4.1 Associated Taxon
1	*Psychrobacter*	*¶ Fusobacterium*	*† Amnipila*
2	** Corynebacterium*	*† Amnipila*	*† Lachnospiraceae_AC2044_group*
3	*JG30-KF-CM45*	*† Catonella*	*† Filifactor*
4	*† UCG-005*	*§ Falsiporphyromonas*	*‡ Fretibacterium*
5	*§ Porphyromonas*	*† Filifactor*	*† Defluviitaleaceae_UCG-011*
6	*† [Eubacterium]_coprostanoligenes_group*	*† Lachnospiraceae_AC2044_group*	*§ Falsiporphyromonas*
7	*§ Bacteroides*	*† Parvimonas*	*† Catonella*
8	*Atopostipes*	*† Mycoplasma*	*† Parvimonas*
9	*§ Prevotella*	*‡ Fretibacterium*	*† Acholeplasma*
10	*† W5053*	*# Treponema*	*† Peptostreptococcaceae*
11	*§ Prevotellaceae_UCG-003*	*† Peptostreptococcaceae*	*# Treponema*
12	** Dietzia*	*† Peptostreptococcus*	*Absconditabacteriales_(SR1)*
13	** Paeniglutamicibacter*	*† Anaerovibrio*	*† Murdochiella*
14	*† Turicibacter*	*† Roseburia*	*† Roseburia*
15	*† Tissierella*	*† Acholeplasma*	*† [Eubacterium]_brachy_group*

** Actinomycetota* (*formerly Actinobacteria*); *† Bacillota* (*formerly Firmicutes*); *§ Bacteroidota*; *¶ Fusobacteriota*; *# Spirochaetota*; *‡ Synergistota*.

## Data Availability

Sequence data were deposited at NCBI Sequence Read Archive (SRA) under accession number PRJNA940284.

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
