# Peer review of "Characterization of the Hoof Bacterial Communities of Active Digital Dermatitis Lesions in Feedlot Cattle"

_microorganisms, 2024, doi:10.3390/microorganisms12071470_

Round 1
Reviewer 1 Report
Comments and Suggestions for Authors
Dear Authors,
the presented study is interesting and scientifically very strong.
I had a few remarks (please see below):
Abstract
The abbreviations in the abstract should be explained
l. 19 an 20: n=12, n=15, n=21: what does it mean?
l. 24-25: why they should be?
l. 27-28: the fact that M2 and M4.1. is finally explained in the last two lines of the Abstract. Should be provided earlier.
Introduction
l. 81-82. Can the Authors provide some more information about the expected outcomes of the examinations? I.e. if the microbiota characterization of different DD stages will provide some particularly important information? Or what will be the implications id the results obtained?
Methods are described adequately and in sufficient details.
Results
Figure 5 is incomprehensible. Both the inner tree and outer ring.
Comments on the Quality of English LanguageThe English language is fine, apart from a few minor errors throughout the text.
Reviewer 2 Report
Comments and Suggestions for Authors
Dear Editors
Thank you for the invitation to review the article "Characterization of the hoof bacterial communities of active digital dermatitis lesions in feedlot cattle article”. Studying and understanding the microbiota is a challenging and interesting topic. However, we need to be careful. The article uses state-of-the-art techniques to answer this question and evaluate specific microbiomes, using and comparing with controlled microbiomes. The metagenomic analysis was done correctly as much as possible. However, the outcome of the results to conclusion must be re-analyzed.
Digital dermatitis (DD) has been reported in feedlots in North America. Authors have already reported in a previous study, evaluating 2,854 cattle confined in 11 stalls in 2 confinements (fortnightly walks) throughout the feeding cycle, the importance of studying this topic. These authors concluded that it is important to maintain good pen conditions to reduce the risk of DD, which can be managed through adequate housing density and strategic bedding, regardless of foot and leg conformation. First question, why didn't the authors study the environment (soil) where these cattle live, for a better association of the microbiota?
In another article, the authors provide a comprehensive characterization of DD and the healthy skin microbiota of feedlot beef cattle and also develop and validate a new multiplex quantitative PCR (qPCR) assay to quantify the distribution of bacterial species associated with DD in different stages of DD injury. Second question, an assay with (qPCR) could help to elucidate the question of the article in question, which is evaluating two specific stages (M2 and M4.1).
These authors quantified the distribution of Treponema, Porphyromonas, Fusobacterium and Bacteroides species in all phases of DD injury. It is worth noting that species were also identified in the article in question. The authors were also able to perform sequencing that revealed that Treponema, Mycoplasma, Porphyromonas and Fusobacterium were associated with DD lesions. Question: Wouldn't it be interesting if the authors of the article in question had this result to make a relatively significant association.
Metagenomics is indeed a cutting-edge technology, however, in some cases, culture is necessary to confirm the etiological agent. In addition to using this state-of-the-art technique, articles also cultivated and identified the main agents associated with DD lesions in biopsy samples, citing the presence of Porphyromonas levii, Bacteroides pyogenes and two Fusobacterium spp. within DD lesions. In previous studies, it was possible to identify agents at the species level; it would be interesting to do this with agents that had a significant relative abundance. Furthermore, the authors themselves cite the importance of Treponema ssp. What species of Treponema would this be in the study in question?
In previous studies, authors reported that early-stage lesions were particularly associated with Treponema media, T. phagedenis and P. levii. There is a lack of identification of Mycoplasma spp. to understand the microbial factors involved in the pathogenesis of DD from this pathogen. These are important gaps and should be explored. The authors of the study in question mention the presence of this agent in the study, but they could look for tools to better explore the presence of this agent.
It is known that previous work, mainly in dairy cattle, has identified several taxo associated with digital dermatitis (DD) lesions and that there is also a gap in the characterization of bacteria at the species level in DD lesions due to database limitations. Knowing this limitation, it is always valid to associate polyphasic identification.
Round 2
Reviewer 2 Report
Comments and Suggestions for Authors
Responding to the letter to the reviewer. I understood the purpose of the article. Please use your own response to the editor and improve the direction of the article "Make this clear in the article". I believe you can improve the discussion. I emphasize again, the discussion is not clear and the authors continue to present results in the discussion. "See line 379, where they cite Table 4". The discussion and moment of clarity where the authors explain the limitations of the study and future perspectives to optimize the question. Based on previous articles.
